# Blood glucose and subcutaneous continuous glucose monitoring in critically ill horses: A pilot study

Valentina Vitale[1], Lise C. Berg[2], Bettina Birch Larsen[2], Andrea Hannesdottir[2], Preben Dybdahl Thomsen[2], Sigrid Hyldahl Laursen[2], Denis Verwilghen[1,2], Gaby van Galen[1,2]*

1 University Teaching Hospital, Sydney School of Veterinary Science, University of Sydney, Camden, New South Wales, Australia, 2 Department of Veterinary Clinical Sciences, Faculty of Health and Medical Sciences, University of Copenhagen, København, Denmark

* gaby@equinespecialists.eu

**Data Availability Statement:** All relevant data are within the paper and its Supporting Information files.

## Abstract

This pilot prospective study reports the feasibility, management and cost of the use of a continuous glucose monitoring (CGM) system in critically ill adult horses and foals. We compared the glucose measurements obtained by the CGM device with blood glucose (BG) concentrations. Neonatal foals (0–2 weeks of age) and adult horses (> 1 year old) admitted in the period of March-May 2016 with clinical and laboratory parameters compatible with systemic inflammatory response syndrome (SIRS) were included. Glucose concentration was monitored every 4 hours on blood samples with a point-of-care (POC) glucometer and with a blood gas analyzer. A CGM system was also placed on six adults and four foals but recordings were successfully obtained only in four adults and one foal. Glucose concentrations corresponded fairly well between BG and CGM, however, there appeared to be a lag time for interstitial glucose levels. Fluctuations of glucose in the interstitial fluid did not always follow the same trend as BG. CGM identified peaks and drops that would have been missed with conventional glucose monitoring. The use of CGM system is feasible in ill horses and may provide clinically relevant information on glucose levels, but there are several challenges that need to be resolved for the system to gain more widespread usability.

## Introduction

Derangements of blood glucose (BG) concentration are relatively common in critically ill horses [1] Hyperglycemia in horses with acute abdominal pain has been linked to non-survival [2, 3], and in neonatal foals hypoglycemia and hyperglycemia occur with approximately equal frequency and are associated with decreased survival [1]. Moreover, a strong positive correlation between plasma L-lactate, a marker of systemic disease severity, and glucose concentrations has been detected both in humans and animals [4] as they are interconnected metabolites in carbohydrate metabolism, and each can lead to the generation of the other [5]. For those reasons, close monitoring of BG concentrations may be useful in critically ill horses

**Funding:** Funding for this research was obtained through the Danish Levy fund (https://hesteafgiftsfonden.dk/).

**Competing interests:** The authors have declared that no competing interests exist.

in order to optimize management and outcome [1]. In critically ill patients, glucose changes in the interstitium commonly develop because of dehydration, fluctuations in local pH, and fluid and protein shifts from the intravascular to the interstitial compartment [6]. As critically ill patients commonly experience insulin resistance and stress-induced alterations in glucose metabolism [7], the combined assessment of blood and interstitial glucose concentration may add additional value to our understanding of the pathophysiological glucose derangement occurring during critical illness.

To avoid repeated blood sampling and subsequent distress for the patient, and to detect peaks and drops in between regular sampling for conventional BG monitoring, devices for continuous glucose measurement (CGM) in the interstitial fluid have been developed and assessed in human medicine [8, 9]. A CGM system stays in place for a prolonged period of time and can thus provide a more complete picture with a higher number of glucose readings and demonstration of glucose trend information in a graphical format [10]. It remains unknown if the changes occurring in the interstitium affect the readings and applicability of CGM systems.

The use of CGM has also been introduced in veterinary medicine [11, 12] with the aim of monitoring glucose concentrations in patients with diabetes or insulin resistance [13]. In horses, the applicability of devices for CGM has been evaluated successfully in healthy exercising horses [14]. In critically ill neonatal foals, CGM has been tested on a small sample size [15], but so far not in adult critically ill horses.

The objectives of this study was, 1) to test and report the feasibility, management and cost of the use of a CGM system in adult horses and foals hospitalized and undergoing intensive care, 2) to compare the glucose measurements obtained by the CGM device with BG concentrations measured by blood gas analysis and a handheld glucometer, and 3) to determine whether the CGM system provided additional clinically relevant information that would remain undetected by conventional glucose monitoring.

## Materials and methods

The study protocol was approved by the ethical committee of Copenhagen University and the Danish Ethical Committee of animal experiments.

### Animals

Patients included in this study were neonatal foals (0–2 weeks of age) and adult horses (> 1 year old) admitted to–blocked for review–in the period of March-May 2016 with clinical and laboratory parameters compatible with systemic inflammatory response syndrome (SIRS). Animals were defined as having SIRS if 2 or more of the following criteria were met: tachycardia (heart rate > 40 beats/minute (adults) and > 120 beats/minute (foals)), tachypnea (respiratory rate > 20 breaths/minute (adults) and > 56 breaths/minute (foals)), hyper- or hypothermia (rectal temperature > 38.5˚C or < 37˚C (adults) and > 39.5˚C or < 37.2˚C (foals)), leukocytosis or leukopenia (> $10x10^9$ cells/L or < $5x10^9$ cells/L (adults) and > $12.5x10^9$ cells/L or < $4x10^9$ cells/L (foals)), presence of > 10% band neutrophils [16].

Informed client consent was obtained prior to study enrollment.

### Data collection

Signalment, diagnosis and treatment of all included subjects was recorded. Blood samples were obtained every 4 hours with a plain syringe and with a blood gas syringe (Safe PICO Aspirator, Radiometer®, Denmark) from an indwelling intravenous catheter inserted in the jugular or cephalic vein after discarding the first 10 ml of blood. The catheter was only used for

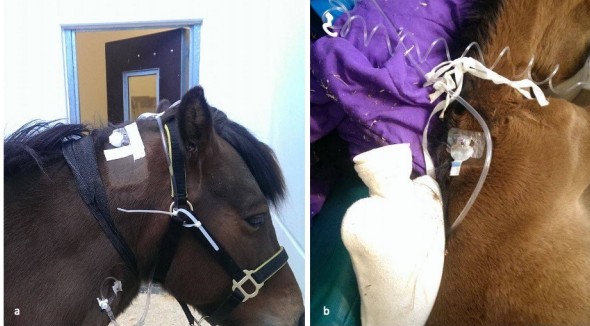

**Fig 1.** Attachment of the CGM system to an adult horse (a) and to a foal (b).

blood sampling and not for the administration of drugs. BG was measured on these samples with a point-of-care (POC) glucometer (OneTouch® Ultra2®, Lifescan IP holdings, LLC) and with a blood gas analyzer (ABL800 Flex blood gas analyzer, Radiometer®, Denmark). The glucometer was used according to manufacturer's instructions and was tested for accuracy at the beginning of the study and every 24 hours using the provided control solution. Blood samples were analyzed within 10 minutes after collection. For every blood sample, packed cell volume (PCV; through centrifuging as microhematocrit), blood lactate and blood pH (as part of the blood gas analysis) were also measured.

The CGM system (Medtronic ENLiTE® glucose sensor, Medtronic MiniMed Northridge, CA 91325 USA) is a portable device consisting of a disposable subcutaneous glucose sensor, a battery-powered transmitter and a battery-powered recording device with a LED monitor. The sensor, a platinum microelectrode coated with a semipermeable membrane, measures $C_{Glu}$-I every 10 seconds. The glucose concentrations are averaged over 5 minutes and then displayed on the monitor. An area of approximately 7x7 cm was clipped on the dorsal part of the neck 10 cm caudally to the ear in adult horses, and 5 cm cranial to the withers in foals. The sensor was inserted in the subcutaneous tissue as previously described [15], fixed with skin glue (Grimas Mastix Watersoluble, PR-nr: 1320912) and covered with an elastic bandage (Fig 1). After a 2-hours initialization period, the device was calibrated based on the BG concentration obtained by the POC glucometer, thus the first measurement obtained with the CGM system from each horse was excluded from statistical analysis, as it represented the first calibration value obtained with the POC glucometer. Subsequently, the POC device was used for calibration 4 hours after initial calibration and then every 12 hours, as advised by the manufacturer. After initial calibration, continuous measurement of $C_{Glu}$-I was started. An alarm was set at 12 mmol/L for hyperglycemia and at 4 mmol/L for hypoglycemia. The device was left in place as long as glucose monitoring was clinically indicated or until the electrode had dislocated or became dysfunctional. Total number of sensors used, number of sensors placed before monitoring could be started, hours passed from the placement of the first sensor and the start of the monitoring, and total hours of monitoring were recorded. Compliance of the patients was also noted, and sensors were checked at least 4 times daily to assess signs of inflammation or infection.

Glucose concentrations were measured during at least 24 hours both in the interstitial fluid ($C_{Glu}$-I) and in the blood ($C_{Glu}$-B).

The total cost per patient was calculated based on the number of sensors used for that specific patient, and the costs as charged by the manufacturer.

## Statistical methods

The results of the CGM system were compared using Bland-Altman analysis to the results of the POC glucometer and to the glucose measurements of the blood gas analysis (BGA), respectively. The percentage of CGM measurements that varied more than 1 mmol/L from BG was determined (a variation of 1 mmol/L or less was decided to be clinically acceptable). The results of the POC glucometer were also compared to those of the BGA. Data of all time points were pooled and analyzed using Bland-Altman analyses.

Furthermore, glucose measurements were categorized into hypo-, normo-, and hyperglycemic events according to BG reference range in adults (4.4–9 mmol/L or 80–160 mg/dL) and foals (4.4–7 mmol/L or 80–130 mg/dL) reported on the blood gas analyzer. Weighted kappa (k) was calculated to assess agreement in classification between the three methods.

Lastly, CGM recordings were compared to blood glucose measurements in all patients to assess whether CGM had detected peaks or drops in glucose that remained otherwise undetected.

## Results

### Study population

Six adult horses (4 mares and 2 geldings; 2 warmblood and 4 ponies) and 4 neonatal foals (3 colts and 1 filly; 3 warmbloods and 1 pony) met the inclusion criteria and were numbered in order of recruitment (case 1, 2, 3. . . and so on). The adult horses had a diagnosis of post-operative ileus following exploratory laparotomy (5/6 cases) and colitis (1/6 cases). The foals presented with neonatal maladjustment syndrome (3/4 cases) and meconium retention combined with SIRS (1/4 cases). One horse was euthanized, and one foal died during the study; all the other animals survived to hospital discharge.

### CGM practicalities

Table 1 shows an overview of the number of sensors used and hours of monitoring for adults and foals. The cost of the CGM device was €588 and the sensors cost €40 per piece, with the costs for the total number of sensors per patient ranging between €40 and €200.

Sensors were removed when they were no longer functional (n = 14), due to death/euthanasia (n = 2), or when the patient was no longer in a critically ill state (n = 2). The sensor insertion and removal and the presence of the CGM system was well tolerated by all patients and there was no evidence of infection or inflammation at the insertion site at any time during or after the study. After initialization and calibration, recordings with the CGM system were successfully obtained in a total of 5 animals (4 adults and 1 foal). In 2 foals the CGM system could not be applied successfully due to excessive movement of the patients resulting in the sensor being pulled out, and no measurements were retrieved. In one adult horse technical problems

**Table 1. Overview of the number of sensors used and recording times.**

| Group of patients | Total number of sensors used until the CGM started (average ± standard deviation; min and max value) | Total number of sensors used (average ± standard deviation; min and max value) | Time from placement of the first sensor until CGM started (average ± standard deviation; min and max value)* | Total time of recordings (average ± standard deviation; min and max value) |
|---|---|---|---|---|
| Adults (n = 6) | 2 ± 1.1; 1–3 | 2.3 ± 1.6; 1–5 | 3.8 ± 1.9; 2.2–6.3 | 35.7 ± 24.1; 1.6–71.5 |
| Foals (n = 4) | 3 ± 1.4; 1–4 | 3.2 ± 1.7; 1–4 | 2.8 ± 2.2; 2.2–3.4 | 10.7 ± 19.7; 0.8–40.2 |

*This includes the mandatory 2 hours of initialization

with the CGM system monitor prevented the recordings. In the remaining cases recordings were successfully retrieved, however not without challenges; defects of the sensors (n = 4), difficulties in inserting the sensor in adults (n = 3) and sensor being pulled out in a foal were encountered.

## CGM versus BG

CGM recordings were available from 5 horses (4 adults and 1 foal), and these were used for statistical comparison between the 3 methods. In total, there were 63 data pairs for comparison of the CGM system to the POC glucometer and BGA, and 69 data pairs for comparison of the POC glucometer to the BGA.

The results of the Bland Altman analyses are listed in Table 2 and summarized for the adult group in Fig 2. The agreement between the POC glucometer and BGA was substantial in all the animals and time point. Overall, fluctuations in interstitial glucose concentrations obtained with the CGM system corresponded fairly well with glucose concentration changes in the blood. In 71.5% of the measurements the variation between interstitial glucose and BG was within the 1 mmol/L limit, but the maximal difference recorded was of 3.7 mmol/L. Interstitial glucose did not always follow the pattern of BG and was sometimes higher but other times lower, with no consistency in the direction within or between different cases. In one patient (case 10), the CGM system measurements were almost consistently lower than the BG concentration. Furthermore, the last measurements from case 4, just before death and with a partial pressure of oxygen (PaO$_2$) of 20.7 mmHg, reported a high discrepancy between blood (7.8 mmol/L) and interstitial (5.1 mmol/L) glucose concentration with complete diversion of the trends of blood (going down) and interstitial glucose (going up; Fig 3). Moreover, in most of the cases there appeared to be a delay of interstitial glucose following the blood glucose trends.

**Table 2. Bland Altman analyses and comparative statistics of the three glucose measurements methods.**

| Cases | CGM system *vs* POC glucometer | | | CGM system *vs* BGA | | | POC glucometer *vs* BGA | | |
|---|---|---|---|---|---|---|---|---|---|
| | Mean bias (95% limits of agreement) (mmol/L) | Max. difference (mmol/L) | % of clinical acceptable measurement differences | Mean bias (95% limits of agreement) (mmol/L) | Max. difference (mmol/L) | % of clinical acceptable measurement differences | Mean bias (95% limits of agreement) (mmol/L) | Max. difference (mmol/L) | % of clinical acceptable measurement differences |
| Case 1 (adult) -19 measurements | 0.04 (-2.3 to 2.4) | 2.3 | 61.1 | -0.5 (-2.7 to 1.7) | 2.7 | 55.6 | -0.5 (-1.4 to 0.4) | 1.0 | 100.0 |
| Case 2 (adult) -9 measurements | 0.8 (-0.4 to 2.0) | 1.8 | 62.5 | 0.5 (-0.5 to 1.5) | 1.3 | 75.0 | 0.2 (-1.0 to 0.7) | 0.9 | 100.0 |
| Case 3 (adult) -11 measurements | 0.1 (-3.4 to 3.7) | 3.5 | 50.0 | -0.2 (-3.9 to 3.6) | 3.3 | 60.0 | 0.4 (-0.6 to 1.4) | 1.2 | 90.9 |
| Case 8 (adult) -11 measurements | -0.3 (-3.2 to 2.6) | 2.9 | 33.3 | -1.1 (-3.5 to 1.2) | 2.8 | 33.3 | -0.9 (-2.0 to 0.3) | 1.8 | 57.1 |
| Case 10 (adult) -14 measurements | -1.4 (-4.5 to 1.7) | 3.4 | 40.0 | -1.3 (-4.3 to 1.1) | 3.7 | 40.0 | -0.3 (-1.2 to 0.7) | 1.1 | 83.3 |
| Adult Group – 64 measurements | -0.2 (-2.8 to 2.4) | 3.2 | 80.0 | -0.5 (-2.9 to 1.9) | 3.7 | 52.8 | -0.3 (-1.6 to 0.9) | 1.8 | 86.7 |
| Case 4 (foal) -6 measurements | -0.5 (-2.7 to 1.7) | 3.2 | 80.0 | -0.1 (-2.1 to 1.8) | 2.7 | 90.0 | 0.4 (-0.3 to 1.0) | 0.9 | 100.0 |

CGM = continuous glucose measurement; BGA = blood gas analysis; POC = point of care.

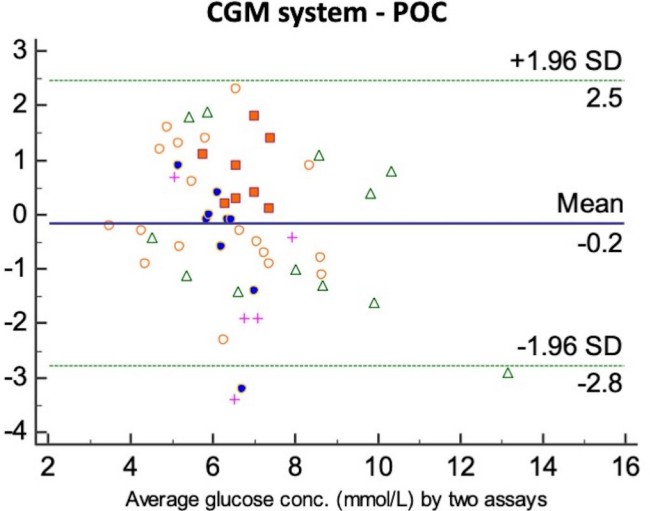

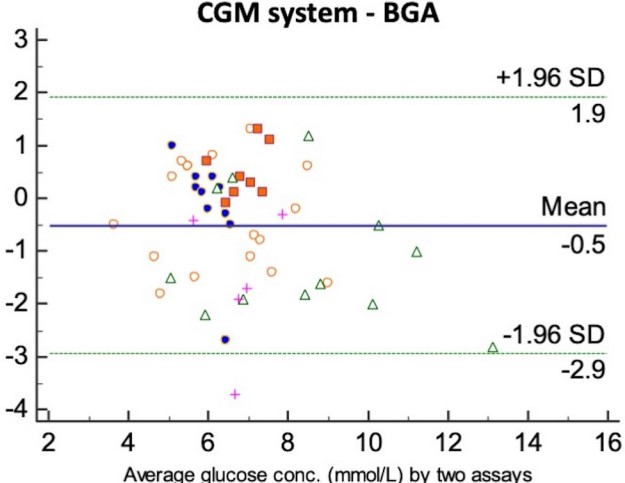

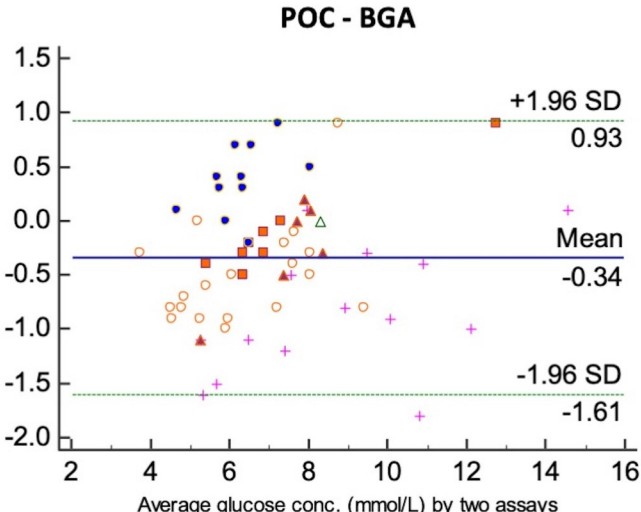

**Fig 2. Bland Altman plots of the adult horse group for the three methods plotted against each other.** The mean of each pair of measurements obtained by two compared methods is plotted on the x-axis, the difference of each pair of these measurements is plotted on the y-axis. The blue midline indicates the mean bias and the two green lines above and below indicate the upper and the lower limits of agreement. Individual cases are represented by a symbol. CGM system = continuous glucose monitoring system; POC = point of care glucometer; BGA = blood gas analysis.

Regarding classification in hypo-, normo- or hyperglycemia in the adult group, agreement was substantial for POC glucometer *versus* BGA (agreement 91.67% of observations, weighted kappa = 0.81) and for CGM system *versus* POC glucometer (agreement in 84.91% of observation, weighted kappa = 0.639); and moderate for CGM system *versus* BGA (agreement in 83.02% of observations, weighted kappa = 0.57).

CGM detected 3 peaks (above 12 mmol/L) and 2 drops (below 4 mmol/L) in glucose levels in a total of 3 patients that were undetected by the 4 hourly BG measurement with POC or BGA. In one of the patients this led to a decision of insulin administration.

## Discussion

The study was performed to investigate the feasibility of using a CGM system in critically ill patients. In the adult group the use of the device was technically feasible, and recordings were obtained, but in foals it was difficult to maintain the sensor in place and obtain a continuous recording due to excessive movement and handling. This made the use of the CGM system difficult in foals in most cases. A previous study using a CGM system in ill foals had more success in obtaining good readings [15]. This difference may be related to a different study population, different management of sick patients, or different placement of the sensor. For more active or neurologic foals such as in the present study, where dislodgement due to movement might be expected, a different location for the sensor may have to be explored in future studies. Hug et al. [15] positioned the sensor either on the neck or on the thorax that, although, are still areas subjected to movement in foals, maybe easier to protect with a bandage compared with the withers. In adult horses, readings were successfully obtained, and sensors remained in place, but other challenges were encountered. First of all, it was difficult to use the provided inserter, and the sensors were found to be more easily placed manually. Another common problem was related to defective sensors, and thus an excessive number of sensors was used per patient (on average 2.33 in the adult group). All of the above issues made it time consuming and expensive to use the CGM system in adult horses.

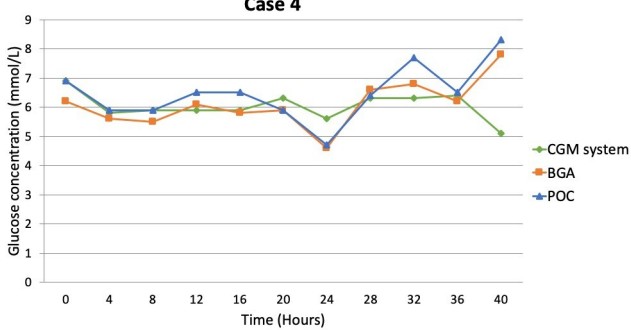

**Fig 3. Recorded glucose concentrations in case 4 with continuous glucose monitoring system (CGM system), point of care glucometer (POC) and blood gas analysis (BGA).**

Our study demonstrated some agreement between BG and CGM values, but there were also differences between blood glucose and interstitial glucose noted, e.g.; 1) delays in fluctuations of interstitial glucose compared to BG, 2) less agreement on patient categorization between CGM *versus* POC and BGA than between POC *versus* BGA, 3) diverting trends between BG and interstitial glucose, and 4) detection of peaks and drops with CGM undetected by POC or BGA.

Fluctuations in blood and interstitial glucose corresponded fairly well and in most cases changes in the interstitial glucose level appeared to be delayed compared to the vascular compartment. Interstitial glucose has been shown to have a time lag of approximately 10 minutes compared to BG under physiological conditions in humans [6]. Under non-physiological conditions such as insulin resistance, hypotension, hypoxia, and high cellular metabolism, the delay can range from 0 up to 40 minutes [17]. Several human studies have found that time delay and interstitial sensor accuracy may vary depending on the patient's condition [9, 18]. This may explain the apparent delay observed in our recordings.

Our results also showed fairly good agreement in patient categorization as being hypo-, normo-, or hyperglycemic between the different methods. However, there was less agreement between CGM versus BGA and POC, than there was between BGA and POC. Hug et al. [15] performed a similar patient categorization in critically ill neonatal foals and described poor agreement between BG and CGM systems, explaining these discrepancies with the systemic hypotension, dehydration, and frequent movements of the foals that might have influenced the accuracy and reliability of the measurements.

The findings of this study support previous studies demonstrating differences between blood and interstitial glucose [6, 19]. In all patients, glucose in the interstitium was sometimes higher and other times lower than blood glucose without a clear pattern. Even though this difference remained within what we believe to be a small margin (1 mmol/L) in the majority of cases (42/70 measurements), in 40% of the measurements it was not, and a maximum difference of 3.7 mmol/L was recorded. We believe that these differences are not necessarily due to method or measurement errors, but more likely reflect physiological reasons and impaired glucose metabolism in severe illness. Measuring interstitial glucose is not a substitute for measuring blood glucose, as they reflect two different compartments [20]. Blood and interstitial fluid differ in a number of aspects with respect to glucose. Blood transfers glucose to all body sites, while interstitial fluid transfers glucose to cells. In one of the patients, this was reflected very clearly. Just before dying and under severe hypoxemia, blood glucose was trending upwards, as would be expected due to increased glycogenolysis and gluconeogenesis stimulated by hormonal changes [21]. At the same time, interstitial glucose was trending downwards. This was believed to be in line with increasing cellular glucose demands in the face of an anaerobic and thus highly inefficient ATP production. Another factor that may play a role in the detection of low interstitial glucose concentration can be the lowering of pH due to metabolic acidosis [21]. In fact, properties of interstitial fluid (pH, temperature and composition) can fluctuate more heavily over time than those of blood [6], especially in the critically ill patient.

CGM systems seem to be relevant for patient care as they provide different, complementary information that can give a more complete insight into the glucose regulatory situation of a given patient [22]. Furthermore, glucose detection based on a single technique is limited by the sensitivity and accuracy of the technique itself. To circumvent this, in human medicine experts advise to use a combination of complementary technique for glucose detection [6]. This was confirmed in this study, where the CGM system provided additional information that was not obtained through the conventional methods for blood glucose measurements showing that fluctuations of glucose in the interstitial fluid did not follow the same trend as

BG, especially in severe illness. Furthermore, to the credit of the continuous monitoring, several clinically relevant peaks and drops in glucose that would have remained undetected with every 4-hour blood sampling, were recorded with the CGM system.

However, when combining the elements of feasibility, the comparison between different glucose measurements, and clinical relevance, CGM systems remain challenging to apply in clinical practice. Also, the cost is much more expensive than the very cheap POC blood glucose measurements, which limits its application further. Nevertheless, it might still have a place for research purposes in order to better define the pathophysiologic mechanisms involved during severe illness, or for specific clinical cases where more a complete insight of glucose metabolism is required.

This study presents several limitations principally related to the small number of cases. Moreover, the SIRS score definition for adult horses is not the last reported by the literature [23] because data were collected prior to the publication of that article and it was not feasible to adapt the cases to the new reference ranges. The measuring range of the device is 2.2–22.2 mmol/L (40–400 mg/dL). Although no values beyond the measuring range were encountered in this study, this should be taken into account for further studies as values beyond this range would be inaccurate. Another limitation of the device is that, although intravascular and interstitial space should be considered as different glucose compartments, the sensor technology requires blood glucose calibrations [9]. This may be a major concern, since sensor calibration during glucose alterations, as those detected in severe illness, may subsequently cause and amplify inaccuracies [24].

In conclusion, the use of CGM systems is feasible in critically ill adult horses. Although we find some technical difficulties with the sensors in neonatal foals, other authors reported that the system was feasible also in this category of patients [15]. Still, there are several challenges related to this device due to the frequent need of changing sensor, the movements of the patient that can cause the sensor to be pulled out, and the cost of the equipment. Furthermore, when glucose homeostasis is compromised, interpretation of the results is not straight forward. Its use may however deepen our understanding of physiological and therapeutic factors that drive glucose homeostasis in critically ill patients. Additional studies are needed to clarify whether there is an actual clinical benefit in terms of reducing morbidities and mortalities of performing CGM in the critically ill patient.

## Supporting information

**S1 Raw data set. Difference between CMGS and POC/acid base clinically acceptable or unacceptable?**
(DOCX)

**S2 Raw data set. Overview–data evaluation–higher / lower.**
(DOCX)

**S3 Raw data set. Overview of % normo-, hypo-, hyperglycemia.**
(DOCX)

**S4 Raw data set. Overview of cases from study start March the 1st until April the 26th 2016.**
(DOCX)

**S5 Raw data set. Overview of sensors.**
(DOCX)

**S6 Raw data set. Pooled data for adult horses (case 1,2,3,8,10).**
(DOCX)

**S7 Raw data set.**
(DOCX)

## Acknowledgments

We gratefully acknowledge the owners of the included patients for allowing us to test this new device, and all clinicians that have contributed to the care of the included animals.

## Author Contributions

**Conceptualization:** Lise C. Berg, Bettina Birch Larsen, Andrea Hannesdottir, Preben Dybdahl Thomsen, Gaby van Galen.

**Investigation:** Bettina Birch Larsen, Andrea Hannesdottir, Sigrid Hyldahl Laursen, Denis Verwilghen, Gaby van Galen.

**Writing – original draft:** Valentina Vitale.

**Writing – review & editing:** Lise C. Berg, Bettina Birch Larsen, Andrea Hannesdottir, Preben Dybdahl Thomsen, Sigrid Hyldahl Laursen, Denis Verwilghen, Gaby van Galen.

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
