## [Decision Letter · Decision Letter 0]

7 Dec 2020

PONE-D-20-35138

Blood glucose and subcutaneous continuous glucose monitoring in critically ill horses: a pilot study

PLOS ONE

Dear Dr. Gaby Van Galen,

Thank you for submitting your manuscript to PLOS ONE. After careful consideration, we feel that it has merit but does not fully meet PLOS ONE’s publication criteria as it currently stands. Therefore, we invite you to submit a revised version of the manuscript that addresses the points raised during the review process.

The article is well written and the scientific purpose and methodologies are deeply described. Minor changes are needed as recommended by reviewer 2

The reviewers are in agreement with the review of the manuscript and minor changes has been proposed.

We look forward to receiving your revised manuscript.

Kind regards,

Arianna Miglio

Academic Editor

PLOS ONE

Additional Editor Comments:

Dear Authors,

the article is well written and deeply described and structured. Please, follow the minor revisions suggested by the reviewers.

Journal Requirements:

2. Please note that PLOS ONE uses a single-blind peer review procedure. We would therefore be grateful if you could include in the information that has been redacted for peer review in the manuscript.

Reviewers' comments:

Reviewer's Responses to Questions

**Comments to the Author**

1. Is the manuscript technically sound, and do the data support the conclusions?

Reviewer #1: Yes

Reviewer #2: Yes

2. Has the statistical analysis been performed appropriately and rigorously? 

Reviewer #1: I Don't Know

Reviewer #2: Yes

3. Have the authors made all data underlying the findings in their manuscript fully available?

Reviewer #1: Yes

Reviewer #2: Yes

4. Is the manuscript presented in an intelligible fashion and written in standard English?

Reviewer #1: Yes

Reviewer #2: Yes

5. Review Comments to the Author

Reviewer #1: This pilot study reports the application of a continuous monitoring system for glucose evaluation in critically ill horses and foals. The glucose concentrations obtained were compared with blood glucose concentration. The idea is interesting and original. Material and methods are consistent with the study. Results reported refer only to one foal and four adult horses, despite the system was applied on more animals. In some of them for causes reported in the results, the system failed. This low number of cases can be accepted for a pilot study, as the values obtained are homogenous and well supported by the explanation in the discussion section. Also the economic impact and costs of this evaluation system are reported and discussed. Study limitations are indicated and discussed

Reviewer #2: The authors’ aim was to verify the feasibility of use a new devise to dose interstitial glucose concentration and the correlation between interstitial and blood glucose (evaluated by PC and blood gas analyzer). The paper is interesting and well structured, but it needs some corrections to be suitable for publication.

Lines 79-84: SIRS in foals: Please you need to use the SIRS score proposed by Roy et al 2017 for adult horse and the one by Wong and Wilkins in 2015 (and 2018 for sepsis) for foals. Please correct and verify if your population can still be included as SIRS positive horses and foals.

I see the data have been collected in 2016, thus maybe you can add tìsome explanation about why you have used a not recent paper for the adult SIRS parameters, but the paper by Wong and Wilkins has been published in the 2015, thus you need to use these parameters for foals’ inclusion in SIRS positive group.

Line 89: was the catheter used only for blood samples? Please add information.

Line 95: what do you mean for “validated”? Please add information.

Line 98: PCV has been evaluated as microhematocrit? Please clarify. After you never discuss this parameter, and this is not relevant for the SIRS inclusion/exclusion. So why did you report this parameter?

Line 113: are the cut-off supported by literature or the cut-off have been chosen by you? Please clarify.

Line 128-129: compared applying what? Bland altman? I think yes (see lines 131-133). If so, in this case you need to better rephrase the sentence in order to have a more understandable and “easy to read” sentence.

Line 131-132: why? This is not one of the aims of the study. In my opinion it should be erased, or you need to change the aim and the title.

Line 136: please add literature for glucose ranges.

Lines 138-139: you say what you did but not haw statistically. Please add.

Table 1: please change “range” with minimum and maximum values. Please add “total number of sensors used until….” in the first row.

Lines 172-173: as my previous comment, this is not one of the aims of the study. In my opinion it should be erased, or you need to change the aim and the title.

Line 174-175: this sentence needs to be moved to mat and met section.

Line 178: fair correlation: both for POC and BGA? You need to specify and discuss both.

Table 2 and Fig. 2. Please erase the comparison between the POC and the blood gas because this is not the aim you stated. The aim was to compare the CGM with BGA or the POC, thus the comparison must be done between CGM vs BGA or POC, not between BGA and POC.

Lines 204-205: again, comparison between BGA and POC is not an aim of your study.

Lines 234 and 247: as previously said.

Lines 261: differs

Line 273: add human patient.

Line 297: in the discussion you said the use of the devise is very difficult in foals, thus this consideration seems to be in contrast with your previous idea. Please clarify or change or erase.

Literature

Corley is 2005 and not 2015.

6. PLOS authors have the option to publish the peer review history of their article (what does this mean?). If published, this will include your full peer review and any attached files.

Reviewer #1: No

Reviewer #2: No

---

## [Author Response · Author response to Decision Letter 0]

8 Feb 2021

Dear reviewers,

Thank you for reviewing our manuscript and providing us with very helpful and constructive feedback. This has allowed us to further strengthen the manuscript. Below we have replied to all your comments. We are looking forward to hear from you again on this manuscript!

kind regards

the authors

The authors’ aim was to verify the feasibility of use a new devise to dose interstitial glucose concentration and the correlation between interstitial and blood glucose (evaluated by PC and blood gas analyzer). The paper is interesting and well structured, but it needs some corrections to be suitable for publication. 

Thank you for the comments. We have changed the manuscript accordingly, please find below the replies to your suggestions.

Lines 79-84: SIRS in foals: Please you need to use the SIRS score proposed by Roy et al 2017 for adult horse and the one by Wong and Wilkins in 2015 (and 2018 for sepsis) for foals. Please correct and verify if your population can still be included as SIRS positive horses and foals.

I see the data have been collected in 2016, thus maybe you can add tìsome explanation about why you have used a not recent paper for the adult SIRS parameters, but the paper by Wong and Wilkins has been published in the 2015, thus you need to use these parameters for foals’ inclusion in SIRS positive group.

This has been adapted to the reference suggested and the citations have been added to the list.

Line 89: was the catheter used only for blood samples? Please add information.

Information has now been added.

Line 95: what do you mean for “validated”? Please add information.

It was tested for accuracy as described accordingly to the manufacturer. The word “validated” has been changed for “tested for accuracy”.

Line 98: PCV has been evaluated as microhematocrit? Please clarify. After you never discuss this parameter, and this is not relevant for the SIRS inclusion/exclusion. So why did you report this parameter?

PCV and other parameters (pH, electrolytes, lactate) where included in the blood gas analysis and performed as routine examination in ICU patients. These values had no influence on the SIRS score but helped to determine when the patient was not in need of glucose monitoring anymore.

Line 113: are the cut-off supported by literature or the cut-off have been chosen by you? Please clarify.

These cut-offs as mentioned were only used to set an alarm on the device. The cut-off used to indicate hypo or hyperglycemia are mentioned later on and are based on the range reported on the blood gas machine.

Line 128-129: compared applying what? Bland altman? I think yes (see lines 131-133). If so, in this case you need to better rephrase the sentence in order to have a more understandable and “easy to read” sentence.

Yes, the sentence has been clarified.

Line 131-132: why? This is not one of the aims of the study. In my opinion it should be erased, or you need to change the aim and the title. 

The results of the glucometer and blood gas analysis were compared to double check the results obtained and to give more strength to the comparison with the continuous glucose monitoring system as it was the only one showing some differences. 

Line 136: please add literature for glucose ranges.

The glucose ranges used where those of our lab.

Lines 138-139: you say what you did but not haw statistically. Please add.

The information has been added.

Table 1: please change “range” with minimum and maximum values. Please add “total number of sensors used until….” in the first row. 

This has been changed.

Lines 172-173: as my previous comment, this is not one of the aims of the study. In my opinion it should be erased, or you need to change the aim and the title. 

The comparison between the glucometer and the blood gas analysis is not the aim of the study this should not be added in the introduction nor in the title. Nevertheless, it supports the data obtained thus need to be mentioned in the results and discussion.

Line 174-175: this sentence needs to be moved to mat and met section.

It has been moved to the section of Materials and methods.

Line 178: fair correlation: both for POC and BGA? You need to specify and discuss both.

As POC and BGA results were almost identical, the comparison with the CGM system and each of them would be the same that’s why it has been summarized in the text, more details can be find in table 2 as mentioned in the results section. 

Table 2 and Fig. 2. Please erase the comparison between the POC and the blood gas because this is not the aim you stated. The aim was to compare the CGM with BGA or the POC, thus the comparison must be done between CGM vs BGA or POC, not between BGA and POC.

Again, although this was not the aim of the study as it has already been done in previous study and the POC as been validated as a reliable method to evaluate the blood glucose concentration in horses, it need to be mentioned to complete and support the data obtained with the CGM system.

Lines 204-205: again, comparison between BGA and POC is not an aim of your study.

As previous replied.

Lines 234 and 247: as previously said.

As previously, this has already been addressed.

Lines 261: differs

Blood and interstitial fluid are two fluids (plural) thus they differ not differs.

Line 273: add human patient.

The phrase is in general, not only in human patients as glucose metabolism physiology in critical illness, although may vary, can still be applied to humans and animals.

Line 297: in the discussion you said the use of the devise is very difficult in foals, thus this consideration seems to be in contrast with your previous idea. Please clarify or change or erase. 

It has been clarified.

Literature

Corley is 2005 and not 2015.

This reference has been deleted.

---

## [Editor Report · Decision Letter 1]

10 Feb 2021

Blood glucose and subcutaneous continuous glucose monitoring in critically ill horses: a pilot study

PONE-D-20-35138R1

Dear Dr. van Galen,

We’re pleased to inform you that your manuscript has been judged scientifically suitable for publication and will be formally accepted for publication once it meets all outstanding technical requirements.

Kind regards,

Arianna Miglio

Academic Editor

PLOS ONE
---

## [Editor Report · Acceptance letter]

15 Feb 2021

PONE-D-20-35138R1 

Blood glucose and subcutaneous continuous glucose monitoring in critically ill horses: a pilot study 

Dear Dr. van Galen:

I'm pleased to inform you that your manuscript has been deemed suitable for publication in PLOS ONE. Congratulations! Your manuscript is now with our production department. 

Kind regards, 

on behalf of

Dr. Arianna Miglio 

Academic Editor

PLOS ONE